# Cross-boundary subsidy cascades from oil palm degrade distant tropical forests

Matthew Scott Luskin [1,2,3], Justin S. Brashares[1], Kalan Ickes[4], I-Fang Sun[5], Christine Fletcher[6], S. Joseph Wright [7] & Matthew D. Potts[1]

Native species that forage in farmland may increase their local abundances thereby affecting adjacent ecosystems within their landscape. We used two decades of ecological data from a protected primary rainforest in Malaysia to illutrate how subsidies from neighboring oil palm plantations triggered powerful secondary 'cascading' effects on natural habitats located >1.3 km away. We found (i) oil palm fruit drove 100-fold increases in crop-raiding native wild boar (*Sus scrofa*), (ii) wild boar used thousands of understory plants to construct birthing nests in the pristine forest interior, and (iii) nest building caused a 62% decline in forest tree sapling density over the 24-year study period. The long-term, landscape-scale indirect effects from agriculture suggest its full ecological footprint may be larger in extent than is currently recognized. Cross-boundary subsidy cascades may be widespread in both terrestrial and marine ecosystems and present significant conservation challenges.

[1] Department of Environmental Science, Policy, and Management, University of California, Berkeley, CA 94720, USA. [2] Center for Tropical Forest Science-Forest Global Earth Observatory, Smithsonian Tropical Research Institute, P.O. Box 37012, Washington, DC 20012, USA. [3] Asian School of the Environment, Nanyang Technological University, 50 Nanyang Avenue, Singapore 639798, Singapore. [4] Department of Biological Sciences, 132 Long Hall, Clemson University, Clemson, SC 29634, USA. [5] Department of Natural Resources and Environmental Studies, National Dong Hwa University, Hualien 97401, Taiwan. [6] Forest Research Institute Malaysia (FRIM), 52109 Kepong, Selangor Darul Ehsan, Malaysia. [7] Smithsonian Tropical Research Institute, Apartado, 0843–03092 Balboa, Republic of Panama. Correspondence and requests for materials should be addressed to M.S.L. (email: mattluskin@gmail.com)

Ecologists have long documented abiotic and biotic exchanges across ecosystem boundaries[1]. The expansion of agriculture and rangelands to nearly half of the Earth's land surface[2] has spurred concerns about spillover effects across cultivated-natural boundaries[1]. For example, ecosystem health in seemingly well-protected reserves is strongly influenced by the surrounding human land uses[3,4]. This is because cultivated areas influence natural ecosystems directly through abiotic edge effects[5], as well as indirectly by modifying interactions between species[6,7]. For example, intensive human food systems can profoundly increase local primary production and the quality of food for wildlife, and can homogenize the location and timing of historically patchy and unpredictable food resources[8]. Animals that travel into farmland to forage can benefit from cross-boundary subsidies, often leading to higher abundances[9,10]. When crop-raiding or livestock-depredating wildlife return to natural habitats and interact with native species, these interactions link the dynamics of cultivated and natural ecosystems[8,11]. Of particular concern are the legacies of ecological winners and losers created by agriculture, as some species thrive through direct or indirect effects of resource subsidies while others suffer[12].

Just as the spatiotemporal availability of agricultural resource subsidies affect ecotone food webs, the movements of subsidized animals can extend the ecological impacts of cultivation into food webs in far away and seemingly unaltered areas[12]. These secondary food-web impacts (cascades) can take many forms, but may ultimately degrade otherwise protected ecosystems (Fig. 1). For example, if a predator benefits from consuming both native and cultivated species, the growth of either prey species can fuel higher local predator abundances (due to either aggregative or demographic responses), which in turn can drive declines in the other prey species[13,14]. This scenario has been shown to indirectly link native and cultivated pest larvae sharing mobile parasitoid wasps[15], native and cultivated plants sharing an herbivore, and livestock and native herbivores sharing a predator[16]. Unfortunately, researching subsidy-driven ecological cascades is often intractable because of difficulties in (i) disentangling and tracking subsidy effects on wildlife demography and movement[1,12] and (ii) identifying and attributing secondary food-web impacts (or 'cascades') to particular agents, especially over large spatial or temporal scales[17,18]. As a result, subsidy cascades have seldom been empirically documented at significant spatial or temporal scales.

Here, we document a case in which the presence of agricultural subsidies increased the reproduction of crop-raiding mobile wildlife, which in turn mediated cascading impacts on the native vegetation of a distant protected area. Specifically, we examined how fallen fruit from oil palm (*Elaeis guineensis*) fueled irruptions of native wild boar (*Sus scrofa*), which then altered the abundance and diversity of understory trees >1 km into a primary forest. Our study was situated in a tropical lowland landscape of Southeast Asia that is a mosaic of oil palm plantations and forest (Supplementary Figs. 1 and 2). Monoculture oil palm cultivation is an ideal system in which to study the ecological impacts of agricultural subsidies because plantations continuously produce fruit for 20–25 years, then are cleared and replanted over a 4- to 6-year period, during which time fruit is completely absent[19]. This creates a natural 'before-after-inference' experiment. Wild boars are a common native generalist species in these forests and their diet is subsidized by abundant oil palm fruit present in nearby plantations (Supplementary Table 1)[20]. Crop-raiding wild boar have great potential as agents of cross-boundary exchange because they travel large distances (>5 km day⁻¹) and create a variety of distinctive soil and vegetation disturbances[21].

We studied wild boar reproduction and forest tree dynamics over two decades in a 130 km² forest reserve in Peninsular Malaysia surrounded by oil palm plantations. Within this forest

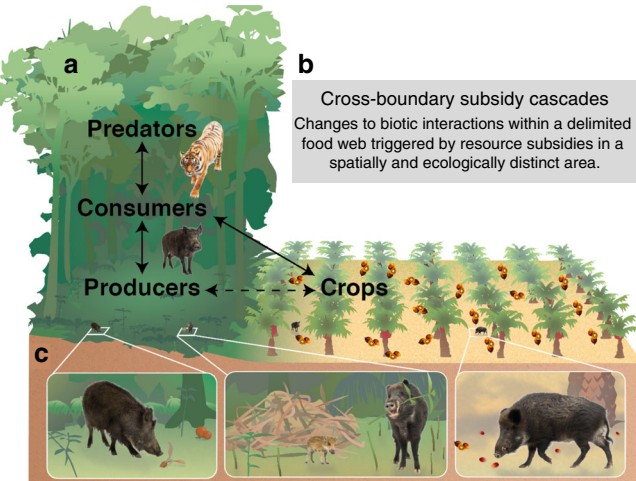

**Fig. 1** Cross-border subsidy cascades from farmland to rainforests. **a** Scene depicting resource subsidies from oil palm plantations consumed by forest wildlife in Malaysia and the secondary trophic effects within forest food webs. Solid lines show direct effects, dotted line shows the indirect effect. This study focuses on the indirect link between native forest trees and cultivated oil palm, mediated by wild boars traveling between habitats. **b** Definition of a cross-boundary subsidy cascade. **c** Scenes depicting crop-raiding wild boar eating forest tree seeds (left), breaking saplings to make nests (middle), and eating oil palm fruits (right)

reserve, we collected data in the Pasoh Research Forest (PRF), a 600-ha core area of primary forest (Fig. 2a). We quantified the existence, scale, and magnitude of the cascading impacts triggered by oil palm subsidies in three steps. First, we determined the direct impact of subsidies on wildlife by quantifying the relationship between oil palm fruit availability (24 years; using standard yield-production models) and wild boar reproduction (19 years; by direct observation of maternal wild boar nests) (Supplementary Fig. 4). Second, we examined whether oil palm-subsidized wild boars caused cross-boundary cascading impacts on tree communities by comparing experimentally fenced wildlife-exclosure plots with open control plots (18 years). Experimental findings provided clear mechanistic evidence that wild boar reproductive activities (breaking tree saplings to build maternal nests) created distinctive size-specific shifts in the understory tree community. Third, we examined whether the distinctive effects identified in the exclosure experiment matched the long-term community-wide shifts in a 50-ha forest dynamics plot (FDP) located immediately adjacent to the experimental plots, which consisted of six tree censuses over 24 years. Importantly, while previous ecological research on forest edge effects has generally focused on changes <600 m[5,17], all our work was done in forest plots located >1.3 km from any non-forest habitat or oil palm plantations.

## Results

**Oil palm fruit subsidies and wild boar reproduction.** Over the 24-year study period, oil palm production underwent two distinct planting cycles and we observed a strong and positive relationship between fruit production and wild boar reproduction in the PRF (Fig. 2b; Supplementary Fig. 4). We began monitoring wild boar during the period between 1995–1998 when the oil palms were mature. Initially, there was a hyper-abundance of wild boar with >300 nests in the 50-ha FDP and density of 27–47 animals per km², as estimated using distance sampling. Later, oil palm subsidies abruptly and nearly completely ceased during the replanting of >95% of the plantations nearby the PRF from 2001

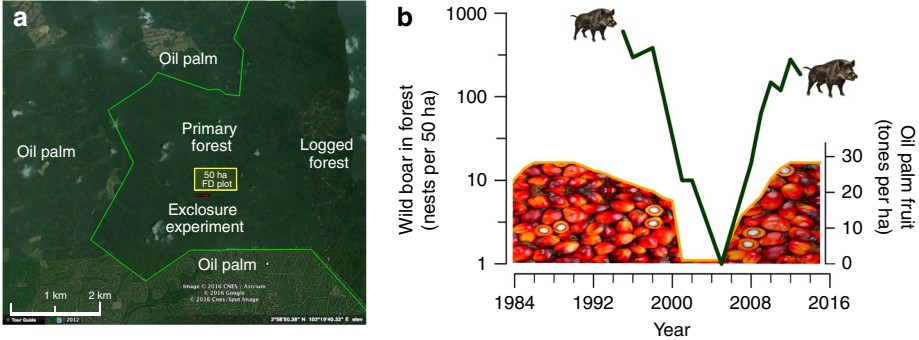

**Fig. 2** Oil palm fruit in plantations drives wild boar irruptions in forests. **a** Pasoh Research Forest and surrounding oil palm plantations (Imagery from Google Earth). All immediately surrounding land use is oil palm plantations, except logged production forest to the east. The forest-plantation edge is outlined in green. Wild boar nest density was monitored in the 50-ha forest dynamics (FD) plot, outlined in yellow (length: 1 km east-west, width: 500 m north-south). The exclosure experiment is shown in red and located 25 m south of the FD plot along the southern plot border. **b** Green line illustrates the number of wild boar nests (left axis) in the Pasoh Research Forest 50-ha FDP as a function of fruit production in oil palm plantations located >1.3 km away. Oil palm plantations were cut down in 2001, replanted in 2003–2004, and began fruiting again in 2006

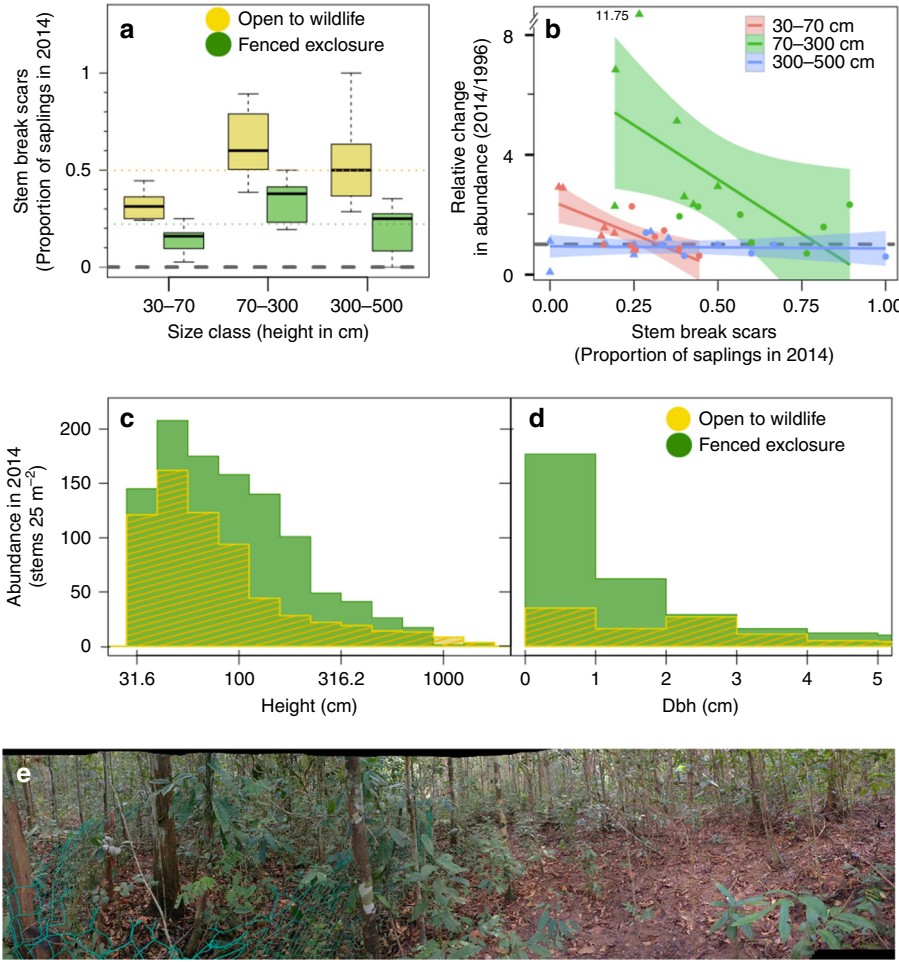

**Fig. 3** Impacts of wild boar disturbance on the forest understory. **a** Different stem break scars in wildlife exclosures (green) compared to open control plots in 2014 (yellow) ($n$ = seven replicates of 25 m$^2$ exclosure plots and paired controls). Colored dashed lines show group means. Whiskers represent distance from upper and lower quartiles to largest and smallest non-outliers. **b** Stem break scars (indicating use in wild boar nests) are associated with a strong reduction in the relative abundance from 1996 to 2014 in small saplings 70–300 cm (green), whereas reductions were smaller for seedlings 30–70 cm (red) and large saplings >300 cm (blue) that are not used in wild boars nests. Shaded polygons show 95% confidence intervals, triangle data points show wildlife exclosures and circles show open controls plots. Y-axis is broken to accommodate an outlier with 11.75 times greater abundance in 2014 compared to 1996. **c** Stem height distribution difference between exclosures and controls in 2014 (Kolmogorov–Smirnov test KS, $d$ = 0.1423, $P$ < 0.001). **d** Dbh size distributions for exclosure and control communities in 2014 (KS, $d$ = 0.29696, $P$ < 0.001). The distinctive shifts wild boar produced on tree stem size distributions were matched in the independent 50-ha FDP dataset (Fig. 4). **e** Photograph of exclosure experiment showing comparatively dense understory in fenced areas (left) vs. open understory in unfenced control plots (right, credit: MSL)

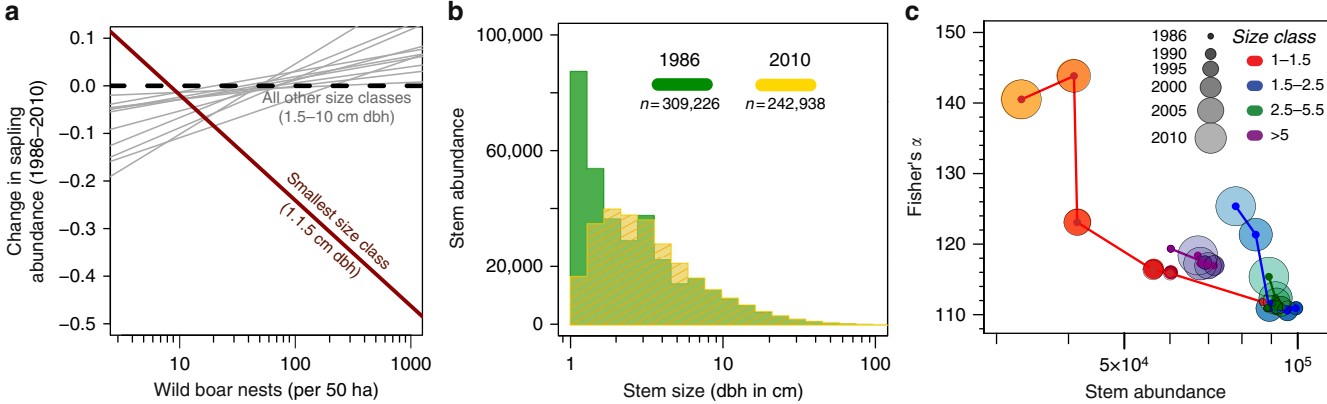

**Fig. 4** Changes in forest tree composition in the 50-ha research plot from 1986 to 2010. **a** Relationship between relative change in abundance of 12 sapling size classes over the five census intervals during which time wild boar nest abundance varied by two orders of magnitude. Gray lines show no significant relationship between wild boar and sapling abundance for twelve sapling size classes (binned by 0.5-cm dbh breaks for stems 1–4 cm dbh and binned by 1-cm breaks for stems 4–10 cm dbh). Red line shows the relationships for the smallest size class (1–1.5 cm dbh; $F_{7,39} = 14.78$, $R^2(adj) = 0.677$, $P < 0.001$). The thick dotted black line indicates no change (null hypothesis). **b** Tree-size distribution shift from 1986 (green) to 2010 (yellow) for all stems within the 50-ha FDP (KS, $d = 0.26109$, $P < 0.001$). **c** Diversity shifts by size class for each FDP census ($n = 435{,}591$ stems for all censuses combined)

to 2006. During those years, nest density dropped 100-fold in the 50-ha FDP. Finally, when oil palm resumed fruiting in 2006 wild boar nest density rebounded 100-fold in the first four years and wild boars were again observed in plantations consuming palm oil fruits each night. Temporal variation in wild boar nest density 1.3–1.9 km into forests was primarily explained by oil palm fruit production in neighboring plantations (linear regression (LR): $F_{1,10} = 28.34$, $R^2_{(adj)} = 0.7131$, $P < 0.001$) (Supplementary Table 2). Alternative explanations fail to account for the marked changes in wild boar nest density; natural food shortages were unlikely because the wild boar population crash of 2001–2006 coincided with three forest-wide "mast fruiting" events in 2001, 2002, and 2005[22], and human hunting and the density of wild boar predators (tigers, *Panthera tigris*, leopards, *Panthera pardus*, and clouded leopards, *Neofelis nebulosa*) did not appreciably increase (Supplementary Table 1). The rapid rate that wild boar nests increased from 2006 to 2012 is within the species' natural reproductive capacity, which is unsurpassed among megafauna[23]. As a result, we believe the overwhelming majority of the additional nests resulted from highly successful in-situ reproduction (demographic response), as opposed to being driven primarily by immigrating wild boars.

**Wild boar effects on tree saplings**. We evaluated whether oil palm subsidies, by increasing wild boar abundance, consequently indirectly caused negative impacts on forest trees in the PRF. We tracked long-term changes in the PRF tree community inside and outside of fenced wildlife exclosures from 1996 to 2014. We established eight 49 m² wildlife exclosures with adjacent paired unfenced control plots in the forest, 1.3 km from the nearest edge. To identify the exact mechanisms by which wild boars affect trees (e.g., seed predation, seedling trampling, or nest-building), we evaluated shifts among three categories of understory trees, each of which has a different interaction with wild boars: seedlings (30–70 cm; commonly damaged by wild boar trampling and rooting[21,24]), small saplings (70–300 cm; broken by pregnant sows to build nests (Supplementary Fig. 3), either causing mortality or leaving a full-circumference scar if the stem resprouts[24]), and large saplings (300–500 cm; too large to be affected by wild boars).

We found that by 2014, the proportion of small saplings with complete stem break scars around their trunks was more than two times higher in open controls (49.88%) vs. exclosures (22.19%)

(likelihood ratio test (LRT): $\chi^2_2 = 17.34$, $P < 0.001$) (Fig. 3a; Supplementary Fig. 5). In addition, small sapling stem abundance was strongly correlated with the proportion of stem break scars (LR: $t_{37} = -3.626$, $P < 0.001$). In contrast, for large saplings, we did not find a significant correlation between abundance shifts and the proportion of stem break scars (LR: $t_{37} = 3.204$, $P = 0.003$; Fig. 3b, Supplementary Fig. 6 and Table 3). More broadly, over the entire 18-year period (1996–2014), sapling abundance decreased 78.38% in controls compared to exclosures (LRT: $\chi^2_2 = 9.6216$, $P = 0.008$), leading to distinctive height and diameter frequency distribution shifts (diameter measured at 1.3 m height, "dbh" hereafter; Fig. 3c, d, Supplementary Fig. 7). These findings reinforce our mechanistic understanding of the impact of wild boar on forest tree communities from our previous work at PRF from 1995 to 1998, during the peak of the wild boar population. During that period, we found that that wild boars removed at least $1602 \pm 516$ saplings per ha per year for their nests, which accounted for 25.4–58.9% of all sapling damage and mortality[24].

**Landscape-scale effects from subsidized wild boar**. Our analyses of the nearby 50-ha FDP revealed subsidized wild boar drove long-term community-wide shifts in the interior forest tree community. The FDP includes large saplings and trees ≥1 cm dbh (trees with dbh of 1cm had a mean height of 249 cm; Supplementary Fig. 7). We found that changes in the abundance of small saplings (1–1.5 cm dbh), which are used in wild boar nests, were negatively correlated with wild boar abundances over the census interval (LR: $t_{39} = -5.095$, $P < 0.001$). However, all other tree-size classes showed no relationship with wild boar abundances (LR: $t_{39} = 1.109$, $P = 0.274$) (Fig. 4a; Supplementary Table 4). This focused disturbance led to a 62.11% reduction in total number of 1–1.5 cm dbh saplings after 24 years and a forest-wide median stem size increase from 2 to 2.9 cm dbh ($n = 309{,}226$ stems; Wilcoxon rank sum test with continuity correction $W = 3.4e^{10}$, $P < 0.001$; Fig. 4b). In contrast to stem density, we found all measures of sapling species evenness increased from 1986 to 2010, with the Fisher's $\alpha$ diversity metric increasing 25% for large saplings 1–1.5 cm dbh (Fig. 4c, $n = 911$ species). This result is consistent with ecological diversity theories wherein preferential removal of dominant species by predators supports increases in rare species (i.e., density-dependent mortality)[24–26]. This disturbance may also shift tree diversity if wild boars are selective

in their harvest of stems or if there are differences in tree species survival or recovery from stem damage.

## Discussion

Subsidy cascades, the indirect change in species interactions driven by a supplemental resource, can be tranferred between distinct ecosystems. We found that the fate of a native tree community in a protected forest was inversely linked to oil palm fruit production in distant agricultural areas through a shared enemy. Wild boars depend on forest habitat for food and safety during daylight hours and the native forest understory provides reproductive female wild boars safe nesting sites and nest materials (tree saplings)[27]. Crop-raiding opportunities on oil palm fruit during evenings attract wild boar to forests in oil palm landscapes (an aggregative population response)[20]. The longer-term demographic response from wild boars crop raiding on oil palm fruit is high reproductive success, which degrades forest understories due to nest building. However, wild boars also disturb the forest understory through direct foraging (e.g., seed predation), trampling, and rooting, which has also facilitated the invasion of a non-native shrub (*Clidemia hirta*) at the PRF[28]. Importantly, the negative impacts also flow back from forests to oil palm because crop-raiding wild boars kill young oil palms[20]. Thus, the irruptions of wild boar that negatively affect oil palm and forests at our study site are contingent on distinct resources provided by each. Wild boar irruptions in forests embedded within oil palm landscapes have been reported in Sumatra[20] and similar processes are likely occurring across wild boar's native and introduced range[23].

We suggest the impacts from cross-boundary subsidy cascades in our region exceed the effects of trophic cascades caused by the loss of predators in other regions[29]. Changes in the PRF were striking, so much so that the forest understory has been conspicuously cleared even to the untrained eye (Fig. 3e). Further, there is evidence that subsidy cascades from human cultivated (or otherwise human-derived foods) may be common globally, and in both terrestrial and marine ecosystems[1]. 'Hot spots' for subsidy cascades include areas with consistent subsidies to mobile species, such as near waste centers[8] (including fisheries bycatch[30]) and where there is direct feeding[10] or baiting[31]. Predators can also be subsidized by livestock depredation[32] and generalist omnivores and mesopredators can be subsidized by any number of human commensal prey species (including pets, pests, or poultry)[33], thus linking domestic prey in human-dominated areas with native prey in natural areas.

Finally, our study emphasizes the extended spatial and temporal scale over which indirect food-web effects from agriculture can degrade distant natural areas. We found strong indirect edge effects over decades in forests >1 km away, suggesting the true global ecological footprint of human food production has been substantially underestimated[17]. With more than 70% of the world's forests occurring within 1 km of an edge[34], subsidy cascades may create significant conservation challenges worldwide. Protecting ecosystems from cascading impacts while still increasing human food production may require nature reserves with larger core areas (e.g., a land-sparing strategy[35]). Easing subsidy cascades in smaller habitat patches may require active management to limit wildlife access to subsidies (e.g., fencing, patrols or lethal management)[20], or 'designer landscapes' with appropriate buffer land uses (e.g., a land-sharing strategy[36]). Reducing wildlife use of food systems may also reduce crop and livestock damage and the risk of zoonotic disease emergence[36]. Moving forward, understanding how to mitigate subsidy cascades will be integral to reconcile food production and nature conservation.

## Methods

**Forest description.** We conducted our research in the 1840-ha Pasoh Research Forest (PRF; 2°58′47′N, 102°18′29′E) located within the 13,000-ha Pasoh Forest Reserve in Negri Sembilan, Malaysia (Supplementary Fig. 2a). The PRF is bordered on three sides by monoculture oil palm plantations that extend 4–10 km in each direction and on the fourth side by the remainder of the actively managed Pasoh Forest Reserve[22] (Supplementary Fig. 2b). Currently, the PRF consists of 1240 ha of forest logged once in 1974 and 600 ha of primary forest. The PRF is managed by the Forest Research Institute of Malaysia (FRIM) and has been a site of nearly continuous tropical forest research since 1975. The 50-ha FDP was established by FRIM in conjunction with the Smithsonian Institute's Center for Tropical Forest Science (now ForestGEO) in 1985[37,38]. The center of the 50-ha FDP is 1.58 km from any forest edge or plantation. The four 50-ha FDP edges range from 1.35 to 1.79 km from any plantations and the mean distance from the plot edge to plantation is 1.48 km. Some have suggested the exclosure experiment was located along the southern edge of the FDP plot, 1.31 km from any forest edge or plantation.

The PRF is a hyper-diverse, aseasonal, humid lowland (80–130 m asl) tropical forest, with a 40–60 m tall canopy dominated by trees in the Dipterocarpaceae family[39]. The first FDP census in 1986 enumerated 335,347 stems belonging to 814 tree species, whereas the 2010 census enumerated 300,211 stems belonging to 922 tree species. Soils are Ultisols on the hills and predominantly Entisols in the flat areas. In addition, this dipterocarp forest like many in the region exhibits general flowering and mast fruiting (GFMF)[40,41]. GFMF occur irregularly at 3–5-year intervals on average and are thought to be triggered by the drought conditions associated with ENSO climatic cycles[41]. During a GFMF, hundreds of species from at least 41 families reproduce synchronously and gregariously. For more details on the ecology of the PRF, see ref.[39]. Some have suggested the removal of oil palm and subsequent wild boar population crash at the PRF in 2001 mimics the natural wildlife crashes observed in non-mast years in the region.

**Wildlife community.** Ninety mammal species (including rodents and bats), 166 bird species, 489 ant species, and 75 reptile and amphibian species have been observed at PRF[37]. Compared to the historic faunal community, large predators and herbivores have been notably absent or greatly reduced in number since the 1980s. Rhinos (*Rhinoceros sondaicus*) and gaurs (*Bos gaurus*) were extirpated in the early 1900s and the last remaining elephants (*Elephas maximus*) were removed by the Malaysian Wildlife Department in 1989 (Supplementary Table 1)[42]. Over the last 30 years, hunting has been light but persistent at PRF, with low densities of forest-dwelling indigenous groups (e.g., the Orang Asli) harvesting species selectively using blow guns, and Malay ethnic groups occasionally hunting muntjac (*Muntjacus muntjak*) and sambar deer (*Rusa unicolor*) within the PRF for meat. As a result, as of 1998, wild boar and the lesser mouse deer (*Tragulus kanchil*) were the only common terrestrial herbivorous or omnivorous species encountered with high frequency[21]. However, since 1998 the wildlife community appears to be recovering, as camera trapping beginning in 2009 has documented leopards (*Pardus neofelis*), Malayan sun bears (*Helarctos Malayan*), tapirs (*Tapirus indicus*), and muntjac.

**Oil palm production.** Production of palm oil around PRF started in the 1970s. The flat lowland areas immediately surrounding PRF were the first areas to be cleared (1970–71). They were then terraced and planted with oil palm between 1976 and 1978[36]. These plantations continuously produced fruit from 1981 to 2001. In 2001, >95% of the plantations within a 2 km buffer surrounding PRF were clear-cut and then replanted in 2003–2004. Fruiting began in 2006 and is ongoing. A GIS study found that in the 60 km × 60 km area encompassing PRF, forest cover decreased from 65.6% in 1976 to 36.3% in 1985, and then to 29.4% in 1996[43]. Over the same period, oil palm plantations increased in the area of extent from 4.9% to 20.6% in the 60 km × 60 km landscape[43]. There was also expansion of other land uses including commonly rubber, rice, bananas, fish farming, and housing (Supplementary Fig. 1).

Malaysia's largest oil palm developer, the Federal Land Development Agency (FELDA), manages the plantations surrounding PRF using practices advocated by the Malaysian Palm Oil Board[19]. Oil palm in all plantations follow a 9 m × 9 m spacing pattern in a triangular formation achieved by offset rows. Although herbicides are periodically applied, groundcover and epiphytes are otherwise left unmanaged unless they became obstacles to harvesting. No riparian areas or "High Conservation Value Forest" are found within the plantations and there is no significant inter-cropping. Plantations at Pasoh reported normal oil palm fruit production for lowland Malaysia[19], and thus we followed Butler et al.[44] to estimate annual fruit production.

**Quantifying wild boar nest abundance.** To track changes in the number of wild boar nests in PFR, we systematically sampled the 50-ha FDP for evidence of wild boar nests in 1995, 1996, 1998, 2001, 2002, 2005, and 2008–2014. Density estimates of wild boar nests were converted to standard estimates in the 50-ha FDP. In 1995, 1996, and 1998, K. Ickes exhaustively recorded all nests present within 25 ha of the FDP[24]. We assessed nest decomposition rates by visiting nests repeatedly over and recorded decomposition status of nests when conducting our sampling. Saplings used in wild boar nests decompose within a year so yearly estimates are considered independent[21]. In 2001, 2002, and 2005 all nests in 50-ha FDP were noted in conjunction with other FDP-wide fieldwork[22,45]. From 2008 to 2014, a

variable number of 6 m × 1000 m transects within the 50-ha FDP transects were conducted by I-F. Sun and an undergraduate field biology course. From 2008 to 2014, the total areas surveyed were 12.44 ha, 6.33 ha, 4.38 ha, 8.38 ha, 3.05 ha, 4.32 ha, and 4.15 ha in successive years.

To determine absolute or relative abundance of wild boar at PFR, K. Ickes conducted distance sampling in 1995–1998[27] and S.J. Wright counted wild boars along three 1.5-km transects (not accounting for distance from observer to animal). The former results are reported in ref. [27] and the latter results are as follows: 120 boars in July–August 2001 when plantations were first being cleared, and four and zero in January–February and September 2002 after plantations were cleared. This report is not presented in Fig. 2 but independently corroborates the timing and magnitude of the wild boar decline during the oil palm rotation. The relationship between oil palm production and wild boar nest density was evaluated using linear regressions, where wild boar nests in the current year were predicted by oil palm production in the current and previous years (Supplementary Fig. 4). When evaluating the relationship between wild boar nests and sapling densities in the 50-ha FDP, the regression results were used to estimate mean nest abundance during the inter-census periods based on oil palm plantation fruit production (Fig. 4a; Supplementary Table 4).

**Fenced wildlife-exclosure experiment**. We constructed eight 49 m$^2$ (7 m × 7 m) open-top exclosures in 1996 that each contained a 25 m$^2$ (5 m × 5 m) vegetation plot, such that there was a 1 m buffer between vegetation plots and fenced areas. Exclosures consisted of 1.5 m tall heavy gauge 4 cm$^2$ mesh metal chain-link fencing anchored by solid wood posts. To ensure that boars did not root under the fence, up to four rows of barbed wire encircled each fence, from ground level to 60 cm in height. This design excluded large (>1 kg) terrestrial animals (e.g., wild boars, tapirs, porcupines, and deer species) but allowed continued access for smaller and arboreal animals (e.g., rodents, civets, primates, birds, and maybe even sun bears). In addition, the open-top design and the wide mesh size minimized changes in microclimate conditions (e.g. altered light, wind speed) within the exclosures. Each exclosure had two paired control plots (25 m$^2$) in a block design. Control plots were established 1 m outside of the exclosures on the two sides that most closely resembled the vegetation structure within the paired experimental plot in 1996.

Woody vegetation in the exclosures and controls was monitored as follows. All trees >30 cm height were tagged, identified and measured between August and September 1996 in all exclosures ($n = 8$, 25 m$^2$ each) and two paired adjacent control plots ($n = 16$, 25 m$^2$ each). The exclosures and control plots were censused a second time in 1998. Seven of the eight exclosures and control plots were recensused a third time in August 2014 (there was severe damage to one exclosure and it was no longer deemed effective at excluding large animals). During the 2014 census, half of each of the both control plots per block were recensused (12.5 m$^2$), totaling 25 m$^2$ per sampling block. A more detailed description of the exclosures experiment design is provided in ref. [23].

We evaluated size-specific changes of seedling and saplings in controls (open to wildlife) and exclosures to determine the causes of long-term changes in the understory tree community. We specifically tested whether wild boar damage was a dominant factor in altering sapling community by quantifying the number of stems that had full-circumference scars, which indicates their previous potential harvest and use in wild boar nests. There was no significant difference in scars between controls and exclosures in 1996 (Wilcoxon signed rank test $W = 23$, $P = 0.8982$), whereas in 2014 there was three times more scars in the open control plots (18.45%) compared to exclosure plots (6.06%) ($W = 4$, $P = 0.007$) (Fig. 3a). To test for size-specific shifts in stem breakage and abundance between treatments we used mixed-effects models and included a random block replicate effect for the seven exclosure-control blocks. Our response variables were proportion of stems with break scars and the proportional change in stems per plot from 1996 to 2014, and we normalized these data by taking the $\log_{10}$ to improve model fit and residuals. We also evaluated if stem breaks were a significant predictor of abundance shifts. We conducted our analyses using the lme4 package in R[46], evaluated model fit using AIC, AICc and the marginal $R^2$[47], all of which can be reproduced by running the code in Datafile S4. To compare the exclosure experiment results with 50-ha FDP, in which only stems larger than 1 cm dbh are measured, we tested if there were changes in the 1–2 cm dbh saplings in exclosures and controls (Supplementary Fig. 6). Matching the 50-ha FDP results, from 1996 to 2014 the total number of tree saplings 1–2 cm dbh rose 58.1% in exclosures and declined by 81.4% in controls.

**FDP analyses**. The FRIM and CTFS-ForestGEO 50-ha FDP has been censused six times (1986, 1990, 1995, 2000, 2005, and 2010). In each census all free-standing woody stems (lianas excluded) >1 cm in diameter at breast height (dbh; 1.3 m) have been mapped (to the nearest 0.5 m) and identified to species. In the initial 1986 census there were 177,725 (53.00% of all stems) in the 1–2 cm dbh "small sapling" size class (including dbh = 1 and 2 cm). Of this initial cohort of small saplings, only 54.68% were alive in the sixth 50-ha FDP tree census (2010). To account for small variations between return times for when stems were measured, we standardized abundance shifts to 5-year periods.

The exclosures established clear mechanistic evidence of how wild boar reduce small saplings densities, but we also considered alternative explanations for the striking decline in the 50-ha FDP. We first evaluated whether progressively lower sampling effort (i.e., uncounted trees) could explain the trend. Different field teams

work through 2-ha sampling units to enumerate all stems in the 50-ha FDP. This independent sampling allows for testing whether trends in the data differed between teams during each census. Trends in the highest density 2-ha units were consistent with declines in the full 50-ha FDP dataset (Supplementary Fig. 7). We next considered whether declining sapling density was normal in the region, and uncovered the opposite trend. Within the 52-ha FDP at Lambir Hills, Malaysian Borneo, which is also situated in a lowland dipterocarp forest adjacent to oil palm plantation, there was consistently higher sapling density from 1992 to 2008[48]. A key difference between these sites may be that Pasoh retains wildlife while Lambir Hills has potentially been defaunated[48]. Finally, to evaluate whether the sapling changes documented at PRF were explained by wild boar, we specifically tested if the smallest class (1–1.5 cm dbh) showed a distinct trend from larger sizes classes that were safe from wild boar disturbances. We posited that background environmental effects such as climate variables, tree fall gaps, or density-dependent recruitment and mortality would act on all small saplings sizes in a similar manner, whereas wild boar effects would only be present in the smallest size class. To account for the spatial autocorrelation of sub-plots within the 50-ha FDP, we analyzed each 50-ha census as a single sample. We used a regression model to control for the effects of density dependence shifts (i.e., based on starting abundances of stems) and temporal autocorrelation. As predicted, the top models (based on AIC, AICc and the marginal $R^2$) included a large significant negative interaction term between mean wild boar nest density and the smallest saplings size class (Fig. 4a; Supplementary Table 4).

**Wild boar population dynamics**. There are clear ecological explanations for the striking magnitude of the wild boar irruptions. First, the decades of consistent fruiting in oil palm plantations represents a stark deviation from the punctuated regional mast fruiting phenology[41,49]. Second, the abundance of fruit: oil palm plantations also produce more fruit (>10 Mg per ha year$^{-1}$) than most tropical forests (<1 Mg fruit per ha year$^{-1}$ on average)[44,49,50]. Third, the dearth of large predators at PRF that naturally consume wild boar may have also been important in allowing wild boar to increase unchecked. However, normal predator territoriality produces densities of 1–2 tigers and 1–5 leopards per 100 km$^{-2}$ in Peninsular Malaysia[51]. Thus, it may be unlikely for these territorial species to obtain high enough abundances to fully suppress oil palm-subsidized wild boar through consumptive or behavioral effects.

**Wild boar control**. There have been a variety of strategies to control wild boars in the region[20]. At PRF, these include limiting access to farmland resource subsidies through fencing, keeping dogs within plantations, and direct lethal management (e.g., hunting, culling). In particular, to reduce wild boar from disturbing newly palm trees, FELDA management spent USD >50,000 to build a 1 m × 1 m trench and install >5 km of metal fencing around their Pasoh plantations. During conversations with M.S.L., FELDA reported this effort was largely ineffective due to storms that filled the trenches with water and falling branches that damaged fences. Wild boars actively swam through flooded trenches and quickly discovered and exploited any fence breaches. Thus, large-scale fencing has logistical, ecological, and effectiveness issues, and may not be a desired component of a landscapes. Alternatively, direct human control can selectively remove crop-raiding animals in plantations[20]. This is particularly feasible where harvested wildlife has locally valuable bushmeat. However, religious and ethnic customs largely shape wildlife consumption and hunting decisions in Southeast Asia, with the Muslim majority population in Peninsular Malaysia adhering to a strict Halal diet that forbids pig meat[20]. The local Chinese minority population does actively hunt wild boar within plantations, but it appears that this hunting pressure has done little to control wild boar populations to date.

**Data availability**. Tree census data is available through the ForestGEO web portal (www.ctfs.si.edu). Wild boar and exclosure data are available in supplementary materials with additional information by request from the authors.

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

## Acknowledgements

ForestGEO program of the Smithsonian Institution provided funding. M.S.L. was supported by a NSF GRFP. Neus Reiolid Balmana helped with the 2014 recensus. S. DeWalt, S. Appanah, S.C. Thomas, R. and J. Ickes, K. Walker, T. Seidler, A. Li, and S. Sahat helped with the initial experimental set up and censusing. We are grateful to E. Gardette and Mohd Sanusi bin Mohamed for help with the exclosure censuses. Artwork in Fig. 1 was created by Judy Jinn. M.S.L. and I-F.S. are thankful to the Director General of the Forestry Research Institute of Malaysia (FRIM) and the Pasoh Research Committee for ongoing permissions to conduct research at Pasoh.

## Author contributions

M.S.L., K.I., and I-F.S. designed the research. M.S.L., K.I., and I-F.S. collected data. M.S.L. analyzed the data. M.S.L., J.S.B., S.J.W., and M.D.P. wrote the manuscript. All authors contributed to the final version of the manuscript.
