## [Peer Review File · Nature Communications]

Reviewers' comments:

Reviewer #1 (Remarks to the Author):

In this study, Luskin and colleagues present data from multiple complementary sources to convincingly make the case that oil palm cultivation subsidizes wild boar populations, which subsequently have a strong negative impact on the recruitment of trees inside protected forests. The data consist of: (a) a wild pig exclusion experiment with detailed tree stand data for the different size classes (b) a time series of wild pig population changes over 24 years, spanning two oil palm production cycles (c) detailed tree stand data for non-experimental areas in the same forest

The mechanisms have been identified in previous papers as early as 2001 (Ickes 2001; Ickes et al. 2005), but the strength of the present manuscript is the synthesis of the different data sources and the length of the time series involved.

The present and anticipated future extent of oil palm cultivation both in Southeast Asia and globally, as well as the generality of the mechanisms demonstrated, mean that this study is of considerable applied interest to a wide range of ecologists, conservationists and agriculturalists worldwide. It is an interesting complement to the "empty forests" issue which emphasizes hunting as a major driver of indirect human impact on forests.

I have no suggestions for major changes.

Two questions though:

Fujinuma and Harrison (2012) make the case that there could be substantial facilitation of the invasive plant *Clidemia hirta* by foraging of wild pigs. This issue is absent of the present study. Why?

Wild pigs may be hyperabundant at Pasoh, but what else did the exclusions exclude?

Detailed comments

L89 "available oil palm fruit nearby forests" is correct but easy to misunderstand – simplify?

L172-173 sentence not quite right, for example "species' selected and mortality" ...?

L280 remove capital in "Melayan"

L286-287 the forest loss is larger than the area planted with oil palm. What happened to the rest?

L299 "in a 25 ha of the FDP" has a word missing?

L321 In "a two paired control plots" delete "a"?

L324 "woody vegetation in the exclosures was monitored..." What about the controls?

L333 Is there a reference for the sapling size classes used to determine the suitability for wild pigs as nest material?

L392 remove plural from "landscapes"?

References

Fujinuma and Harrison 2012 <https://doi.org/10.1371/journal.pone.0037321>

Ickes 2001 [http://dx.doi.org/10.1646/0006-3606\(2001\)033\[0682:HAONWP\]2.0.CO;2](http://dx.doi.org/10.1646/0006-3606(2001)033[0682:HAONWP]2.0.CO;2)

Ickes et al. 2005 <http://onlinelibrary.wiley.com/doi/10.1890/04-0867/full>

Reviewer #2 (Remarks to the Author):

Luskin et al. review

I think this manuscript addresses a novel and interesting topic, and the evidence assembled that oil palm cultivation influences damage and dynamics of trees in nearby 'protected' forest is

compelling. The suggestion that similar indirect effects may play out in other systems where shared enemy numbers are boosted by human-mediated resource subsidies will make this work of wide interest.

I do think the authors need to do more to convince the reader that they are documenting "apparent competition" and they should consider carefully whether it is appropriate to frame the study in terms of apparent competition mediated by a shared herbivore. Citations to the theory and principles of apparent competition (e.g. Holt's original 1977 coining of the concept, and later papers by Holt & Lawton, Holt & Kotler and other) are lacking, as is any consideration of the sorts of evidence that allows apparent competition to be distinguished from other forms of indirect interaction and considerations about its symmetry or asymmetry (e.g. Chaneton & Bonsall 2000). Without this, I think the title and framing within the manuscript should be in terms of plantation resource subsidies having spillover effects in nearby natural habitats, rather than apparent competition per se.

There are two types of apparent competition: short term and long-term, where natural enemies (here: wild boar) respond to resources as an aggregative response or a demographic response, respectively. I think this distinction would be really useful here, as I was left wondering whether the trends in wild boar abundance observed are a consequence of demographic responses to oil palm fruit availability (high population growth rates) or aggregative responses involving movement of wild boar individuals into the study area to take advantage of newly-available resources.

What would be 'natural' wild boar densities i.e. in the absence of humans? Could it be that high (and fluctuating: see Curran, L. M., & Leighton, M. (2000). *Ecological Monographs*, 70, 101-128) wild boar densities are actually a natural component of forest dynamics, with this impact restored by the presence of oil palm rather than removed?

While I value the study greatly, the presentation is marred by a large number of poorly phrased sentences and typographical errors. As a sample:

Line 172 – "Further, this disturbance may also cause shifts in diversity due _to_differences between _species_ selected and mortality versus resprouting of harvested stems".

Line 356 – "Abundance shift during census intervals were standardized to 5-yr periods using mean interval between when stems were measured.". This mangled sentence needs to be re-written

Line 362 "declining sapling density" not "declining saplings density"

Line 364 "higher sapling" not "higher in sapling".

Line 468 "This figure is meant to aid..." – very confusing phrasing.

Supp T1 – "Neofeis" should be "Neofelis", *Presbytis obscura* should be italicized

Line 502 – "performed", not "preforned"

There are plenty of others but I ran out of enthusiasm for listing them. The whole manuscript would benefit from a thorough and critical read-through to remove these and improve clarity.

Additional comments:

Abstract: "Here we provide rare empirical evidence of the indirect food web impacts from anthropogenic resource subsidies to mobile native wildlife being transferred to distant pristine ecosystems." This is a key sentence but I found it confusing and had to re-read 3 times to get the intended meaning. Re-phrase.

Line 59 – the Phalan et al. reference (on sharing/sparing) doesn't seem relevant in the context cited; it is about 'winners' and 'losers' but more in the context of intensification than agricultural provisioning of resources.

Line 62 – not really a food web study. Better to say 'more distant natural communities'.

Line 64 – needs a clear definition of apparent competition and citation to the original literature on apparent competition (Holt).

Line 96 – ‘oil palm subsidy-triggered cascading impacts’ Re-phrase to avoid confusing list of four adjectives before a noun, e.g. ‘cascading impacts triggered by oil palm subsidies.’

Line 101 - ‘unmanipulated plots’ ?

Figure 1 legend - ‘forest-dwelling forest wildlife’ – re-phrase

Line 343 – R is the software. CRAN (not Cran) is the network of servers hosting it

Line 378 – Citations have been merged as “234748378”.

Reviewer #3 (Remarks to the Author):

Luskin et al provide a solid, long-term data set examining the indirect food web impacts that occur when native animals utilize the resources provided by large-scale agriculture. In this case, they track the effects of food resources made available in oil palm plantations on wild boars which in turn have a measurable and substantial impact on woody plants density in native forests surrounded by cropland.

The data collection and analysis are rigorous and well executed. The study is aided by the presence of a 24-yr data set from a forest dynamics plot located adjacent to the experimental plots that enables the authors to examine long-term community effects. Overall, the paper is present a fairly novel documentation of spillover effects from agricultural lands to native forests and, as such, should be of interest and value to conservation and land management. This paper will no doubt spark further studies into such relationships across other agricultural landscapes.

One item that is conspicuously absent is some mention of how these dynamics may play out in areas where *Sus scrofa* (an invasive species with huge impacts on native ecosystems globally) have been introduced. Agricultural subsidies led to increase wild boar populations and detrimental effects on native forests in a location where they are native. Looking forward, can the authors suggest how this study might relate to management of the agriculture/natural area interfaces across the extensive areas where these animals are feral?

NCOMMS-17-08684 Response to reviewers

Reviewer comments in black, responses in blue text.

Reviewer #1 (Remarks to the Author):

General comments

In this study, Luskin and colleagues present data from multiple complementary sources to convincingly make the case that oil palm cultivation subsidizes wild boar populations, which subsequently have a strong negative impact on the recruitment of trees inside protected forests. The data consist of: (a) a wild pig exclusion experiment with detailed tree stand data for the different size classes (b) a time series of wild pig population changes over 24 years, spanning two oil palm production cycles (c) detailed tree stand data for non-experimental areas in the same forest

The mechanisms have been identified in previous papers as early as 2001 (Ickes 2001; Ickes et al. 2005), but the strength of the present manuscript is the synthesis of the different data sources and the length of the time series involved.

The present and anticipated future extent of oil palm cultivation both in Southeast Asia and globally, as well as the generality of the mechanisms demonstrated, mean that this study is of considerable applied interest to a wide range of ecologists, conservationists and agriculturalists worldwide. It is an interesting complement to the “empty forests” issue which emphasizes hunting as a major driver of indirect human impact on forests.

I have no suggestions for major changes.

Two questions though:

Fujinuma and Harrison (2012) make the case that there could be substantial facilitation of the invasive plant *Clidemia hirta* by foraging of wild pigs. This issue is absent of the present study. Why?

We are grateful for the referee’s overall support and agree this is an important point. We have added the following sentence and reference to the discussion at line 239: *At our site, wild boar soil disturbances have also been shown to facilitate the invasion of a non-native shrub (Clidemia hirta)³².*

Wild pigs may be hyperabundant at Pasoh, but what else did the exclusions exclude?

A valid concern that is now discussed more clearly in the Methods section “Exclosure” section Community” (lines 346-349):

This design excluded large (>1 kg) terrestrial animals (e.g. wild boars, tapirs, porcupines, and deer species) but allowed continued access for smaller and arboreal animals (e.g. rodents and sun bears).

Detailed comments

L89 “available oil palm fruit nearby forests” is correct but easy to misunderstand – simplify?

We have changed the sentence to read: *Wild boars are a common native generalist species in these forests and their diet is subsidized by abundant oil palm fruit present in nearby plantations (Supplementary Table 1)²¹.*

L172-173 sentence not quite right, for example “species’ selected and mortality” ...?

We have clarified this sentence.

L280 remove capital in “Melayan”. Corrected.

L286-287 the forest loss is larger than the area planted with oil palm. What happened to the rest?

This had been clarified in the revision (line 314):

Over the same period, oil palm plantations increased in extent from 4.9% to 20.6%⁴⁵. There was also expansion of other land uses including commonly rubber, rice, bananas, fish farming, and housing

L299 “in a 25 ha of the FDP” has a word missing? Corrected.

L321 In “a two paired control plots” delete “a”? Corrected.

L324 “woody vegetation in the exclosures was monitored...” What about the controls?

Thanks for pointing this out, the sentence now reads:

Woody vegetation in the exclosures and controls was monitored as follows.

L333 Is there a reference for the sapling size classes used to determine the suitability for wild pigs as nest material? Our previous work (by K. Ickes) investigated this in two papers. We have added the appropriate reference.

L392 remove plural from “landscapes”? Corrected – thank you!

Reviewer #2 (Remarks to the Author):

General comments

I think this manuscript addresses a novel and interesting topic, and the evidence assembled that oil palm cultivation influences damage and dynamics of trees in nearby ‘protected’ forest is compelling. The suggestion that similar indirect effects may play out in other systems where shared enemy numbers are boosted by human-mediated resource subsidies will make this work of wide interest.

I do think the authors need to do more to convince the reader that they are documenting “apparent competition” and they should consider carefully whether it is appropriate to frame the study in terms of apparent competition mediated by a shared herbivore. Citations to the theory and principles of apparent competition (e.g. Holt’s original 1977 coining of the concept, and later papers by Holt & Lawton, Holt & Kotler and other) are lacking, as is any consideration of the sorts of evidence that allows apparent competition to be distinguished from other forms of indirect interaction and considerations about its symmetry or asymmetry (e.g. Chaneton & Bonsall 2000). Without this, I think the title and framing within the manuscript should be in terms of plantation resource subsidies having spillover effects in nearby natural habitats, rather than apparent competition per se.

We appreciate the reviewer’s positive, constructive and thorough review. Following the reviewer’s advice, we have re-focused the Introduction and Discussion away from apparent competition, instead focusing on the role of food subsidies in triggering cascading impacts. We hope our new easy phrasing (‘subsidy cascades’) will be helpful to describe this widespread issue, especially since subsidy research is rapidly increasing within the fields of ecology and conservation biology.

There are two types of apparent competition: short term and long-term, where natural enemies (here: wild boar) respond to resources as an aggregative response or a demographic response, respectively. I think this distinction would be really useful here, as I was left wondering whether the trends in wild boar abundance observed are a consequence of demographic responses to oil palm fruit availability (high population growth rates) or aggregative responses involving movement of wild boar individuals into the study area to take advantage of newly-available resources.

This is a very important and interesting distinction that we have now included in the Intro, Results, and Discussion. Specifically, the following sentences have been added (line 71):

For example, if a predator benefits from consuming both native and cultivated species, the growth of either prey species can fuel higher local predator abundances (due to either aggregative or demographic responses), which in turn can drive declines in the other prey species^{15,16}.

Line 146:

The rate that wild boar nests increased from 2006-2012 is within the species' natural reproductive capacity, which is unsurpassed among megafauna²⁵. As a result, we believe the overwhelming majority of the additional wild boar nests resulted from highly successful in-situ reproduction (demographic response), as opposed to being primarily from immigrating females.

Line 339:

Crop-raiding on oil palm fruit during evenings attracts wild boar to forests in oil palm landscapes (an aggregative population response)²². The longer-term demographic response from oil palm subsidies is high wild boar reproductive success, which degrades forest understories due to nest building.

What would be 'natural' wild boar densities i.e. in the absence of humans? Could it be that high (and fluctuating: see Curran, L. M., & Leighton, M. (2000). Ecological Monographs, 70, 101-128) wild boar densities are actually a natural component of forest dynamics, with this impact restored by the presence of oil palm rather than removed?

This is an interesting point. We have added an appendix to discuss this issue and the following sentence to line 279: *The removal of oil palm and subsequent wild boar population crash at the PRF in 2001 thus mimics the natural crashes observed in non-mast years in the region (Appendix 1).*

While I value the study greatly, the presentation is marred by a large number of poorly phrased sentences and typographical errors. As a sample:

Line 172 – “Further, this disturbance may also cause shifts in diversity due _to_ differences between _species_ selected and mortality versus resprouting of harvested stems”.

Our apologies for the errors. **This is now corrected.**

Line 356 – “Abundance shift during census intervals were standardized to 5-yr periods using mean interval between when stems were measured.” This mangled sentence needs to be re-written. **Corrected.**

Line 362 “declining sapling density” not “declining saplings density”. **Corrected.**

Line 364 “higher sapling” not “higher in sapling”. **Corrected.**

Line 468 “This figure is meant to aid...” – very confusing phrasing. Corrected.

Supp T1 – “Neofeis” should be “Neofelis”, *Presbytis obscura* should be italicized

Line 502 – “performed”, not “preforned”, Corrected.

There are plenty of others but I ran out of enthusiasm for listing them. The whole manuscript would benefit from a thorough and critical read-through to remove these and improve clarity.

We apologize for these additional errors. We have undertaken more careful proofreading and hope most other errors were corrected.

Additional comments:

Abstract: “Here we provide rare empirical evidence of the indirect food web impacts from anthropogenic resource subsidies to mobile native wildlife being transferred to distant pristine ecosystems.” This is a key sentence but I found it confusing and had to re-read 3 times to get the intended meaning. Re-phrase.

Thank you, we have edited the text to read:

Here, we provide empirical evidence that agriculture subsidizes native wildlife that degrade pristine ecosystems.

Line 59 – the Phalan et al. reference (on sharing/sparing) doesn’t seem relevant in the context cited; it is about ‘winners’ and ‘losers’ but more in the context of intensification than agricultural provisioning of resources.

We agree and removed this citation.

Line 62 – not really a food web study. Better to say ‘more distant natural communities’.

The suggested change was made.

Line 64 – needs a clear definition of apparent competition and citation to the original literature on apparent competition (Holt).

We agree and the changes are described in the previous response above.

Line 96 – ‘oil palm subsidy-triggered cascading impacts’ Re-phrase to avoid confusing list of four adjectives before a noun, e.g. ‘cascading impacts triggered by oil palm subsidies.’

This has been changed to read:

We quantified the existence, scale, and magnitude of the cascading impacts triggered by oil palm subsidies in three steps.

Line 101 - ‘unmanipulated plots’ ?

Changed to ‘open control plots’.

Figure 1 legend - ‘forest-dwelling forest wildlife’ – re-phrase

Corrected to read “forest wildlife”

Line 343 – R is the software. CRAN (not Cran) is the network of servers hosting it

Corrected.

Line 378 – Citations have been merged as “234748378”.

Corrected.

Reviewer #3 (Remarks to the Author):

Luskin et al provide a solid, long-term data set examining the indirect food web impacts that occur when native animals utilize the resources provided by large-scale agriculture. In this case, they track the effects of food resources made available in oil palm plantations on wild boars which in turn have a measurable and substantial impact on woody plants density in native forests surrounded by cropland.

The data collection and analysis are rigorous and well executed. The study is aided by the presence of a 24-yr data set from a forest dynamics plot located adjacent to the experimental plots that enables the authors to examine long-term community effects. Overall, the paper is present a fairly novel documentation of spillover effects from agricultural lands to native forests and, as such, should be of interest and value to conservation and land management. This paper will no doubt spark further studies into such relationships across other agricultural landscapes.

One item that is conspicuously absent is some mention of how these dynamics may play out in areas where *Sus scrofa* (an invasive species with huge impacts on native ecosystems globally) have been introduced. Agricultural subsidies led to increase wild boar populations and detrimental effects on native forests in a location where they are native. Looking forward, can the authors suggest how this study might relate to management of the agriculture/natural area interfaces across the extensive areas where these animals are feral?

We appreciate the reviewer's enthusiasm for our study. Following her/his suggestion, we have added the following sentence in the Discussion (lines 234-236) and citation (25) that describes the many impacts of wild boar globally:

Wild boar eruptions in forests embedded within oil palm landscapes have been reported in Sumatra²² and similar processes are likely occurring across wild boar's native and introduced range²⁵.

We would like to discuss this in more detail but have run into our word limit.

REVIEWERS' COMMENTS:

Reviewer #1 (Remarks to the Author):

The authors have taken care of all the comments in a satisfactory manner, I find. As stated in my previous review, this is a very interesting and important study.

Only one thing caught my eye: I note that one of the author's names has a typo (Flectcher should be Fletcher I think)

Reviewer #2 (Remarks to the Author):

The authors have addressed my comments and those of the other reviewers thoroughly and effectively, and I think this really interesting manuscript is now ready for publication. The authors may wish to acknowledge the input of the three anonymous reviewers.

Reviewer #3 (Remarks to the Author):

The authors adequately addressed the concerns expressed and I find the manuscript in good condition for publication.